# Emergent honeycomb network of topological excitations in correlated charge density wave

Jae Whan Park[1], Gil Young Cho[2], Jinwon Lee[1,2] & Han Woong Yeom ● [1,2]

When two periodic potentials compete in materials, one may adopt the other, which straightforwardly generates topological defects. Of particular interest are domain walls in charge-, dipole-, and spin-ordered systems, which govern macroscopic properties and important functionality. However, detailed atomic and electronic structures of domain walls have often been uncertain and the microscopic mechanism of their functionality has been elusive. Here, we clarify the complete atomic and electronic structures of the domain wall network, a honeycomb network connected by $Z_3$ vortices, in the nearly commensurate Mott charge-density wave (CDW) phase of 1T-TaS$_2$. Scanning tunneling microscopy resolves characteristic charge orders within domain walls and their vortices. Density functional theory calculations disclose their unique atomic relaxations and the metallic in-gap states confined tightly therein. A generic theory is constructed, which connects this emergent honeycomb network of conducting electrons to the enhanced superconductivity.

[1] Center for Artificial Low Dimensional Electronic Systems, Institute for Basic Science (IBS), 77 Cheongam-Ro, Pohang 790-784, Korea. [2] Department of Physics, Pohang University of Science and Technology, Pohang 790-784, Korea. Correspondence and requests for materials should be addressed to H.W.Y. (email: yeom@postech.ac.kr)

Competing periodicities and discommensuration phenomena have been widely discussed from one-dimensional (1D) chain models[1–3] to surface adsorbate monolayers[4,5] and more recently 2D van der Walls stacking of atomic crystals[6–8]. For example, the commensuration interaction in graphene layers induces various superstructures of discommensuration with exotic electronic property such as unconventional superconductivity[9,10]. Discommensurations are also popular and important in 2D systems with more complex interactions, such as charge ordered or charge-density wave (CDW) systems, where domain wall networks have been suggested crucial in metal–insulator transitions[11,12], emerging superconductivity[13–15] and the formation of various metastable states for device functionality[16–18].

Among diverse 2D materials, exotic low-dimensional electronic properties and domain walls of 1T-TaS$_2$ CDW are outstanding[19–21], as combined further with the strong electron correlation[22] and the strong spin frustration[23,24]. 1T-TaS$_2$ is composed of layers of hexagonal tantalum atoms coordinated octahedrally by sulfur atoms. A metallic phase at high temperature transits into an incommensurate CDW phase below 550 K, a nearly commensurate CDW (NC-CDW) below 350 K, and a commensurate CDW (C-CDW) below 180 K[11]. The atomic structure of the C-CDW phase has the unit of the well-known David-star cluster; among 13 Ta atoms of a cluster 12 are distorted toward the central one forming a $\sqrt{13} \times \sqrt{13}$ supercell[25]. One unpaired electron in the central Ta atom falls into the Mott-insulator state by onsite Coulomb repulsion[22]. The NC-CDW phase was reported to consist of C-CDW domains and a honeycomb domain-wall network[26–31] with its atomic structure still unknown and its electronic property under debates[13,14,16]. Recent experiments reported that various metallic excited states are formed, presumably with highly conducting domain wall networks, by pressure, doping and other control parameters[16,17,32–37]. These states may be utilized in a ultrafast memory[16] or a memristor device[17]. The domain walls are also suggested as the origin of superconductivity emerging from those metallic states[13,14,38,39]. However, recent STM experiments observed the nonmetallic nature of domain walls for the voltage-pulse-induced domain-wall phase[33,34] and isolated domain walls at low temperature[40]. Under these contradictory findings, the characterization of atomic and electronic properties of domain walls in the NC-CDW phase and the general microscopic understanding of the role of the domain wall network for emerging quantum phases and functionalities are highly requested.

In this work, we directly resolve atomistic details of discommensuration domain walls of the NC-CDW phase in 1T-TaS$_2$ by scanning tunneling microscopy (STM). By comparison with the experiment, our density functional theory (DFT) calculations identify that the domain wall consists of particular types of imperfect David-star clusters. Domain walls are further connected by $Z_3$ topological vortices of domain walls to form a honeycomb network. We also identify the atomic structure and the novel charge order within these vortex cores. The metallic states of the NC-CDW phase originate from imperfect David stars of domain walls and vortices. A generic theory for the conducting honeycomb network indicates symmetry-protected Dirac bands, quadratic band touchings, and flat bands, which would definitely play important roles in the emerging superconductivity. This work provides unprecedented microscopic understanding of discommensurations in 2D materials with strong interactions, which has wide implication into diverse 2D materials with open possibilities for designing functionalities and quantum phases through the discommensuration engineering.

## Results
**Hexagonal lattice of domains**. Figure 1a shows an STM image of the NC-CDW phase in 1T-TaS$_2$ at room temperature. It consists of the domains with relatively high CDW amplitude separated by the low-amplitude regions of domain walls. A hexagonal array of domains and CDW maxima within each domain are more clearly seen in the low-pass filtered image of Fig. 1b. A bright protrusion within a domain corresponds to a single David-star cluster of the C-CDW phase, as revealed more clearly below. The Fourier transform of the STM image (Fig. 1c) exhibits main peaks at $\mathbf{q}_{CDW} = 0.519$ Å$^{-1}$ (12.11 Å) in consistency with the periodicity of C-CDW[21]. In addition, the main peaks carry satellites[41,42] rotated slightly with respect to the C-CDW vector (see the enlargement)[43,44], which have $\mathbf{q}_{HEX} = 0.083$ Å$^{-1}$ (76.13 Å) with a rotation of $9.5 \pm 2°$ from the C-CDW peaks. This represents the hexagonal lattice of domains and a honeycomb domain-wall network.

**Discommensurate domain wall**. An atomic resolution STM image reveals the misalignment of David star CDW clusters between domains. Two adjacent domains with bright protrusions in Fig. 1d are separated by the domain-wall region with less bright protrusions. In the domain wall, David-star clusters from two domains, for example, green and yellow ones, are misaligned to share their two outer Ta atoms (see dashed stars). This domain-wall configuration is the first-type domain wall (DW-1) structure with a CDW translation of $-b$ ($a \times b$ is the inplane unitcell of the undistorted structure) among the 12 possible configurations due to 13 Ta atoms within one C-CDW unitcell as classified previously[33,34]. This particular configuration is found ubiquitously along all hexagonal directions between domain centers over the whole area probed. In the domain area, a single David star exhibits mainly three bright spots arranged in an inverted triangle. They correspond to top three S atoms at the center of a David star cluster. In contrast, major atomic protrusions in the domain wall are arranged mostly in regular triangles (see the enlarged image in Fig. 2c), indicating different types of clusters. The inset of Fig. 1d shows the arrangement of David-star clusters for an ideal hexagonal NC-CDW model as constructed from the STM image. The distance from one domain center to another is 77.47 Å ($22a + 2b$) with about 19 perfect David-stars for each domain and with the angles with respect to the C-CDW vector of 9.6°. These structural parameters agree well with the present and the previous experiments[26]. The center position of David-star clusters in each domain is shifted[28] and domains are connected with the DW-1 structure, which is the key atomic structure of the NC-CDW phase. Note also that this structure has regions where three domain walls meet (the uncolored David star clusters in Fig. 1d). They are vortices of domain walls as discussed further below.

**Atomic structure of domain wall**. In order to identify the atomic structure of this characteristic domain wall, we performed DFT calculations. A simplified supercell of a reduced size from the ideal structure of Fig. 1d is used, which consists of five perfect David-star clusters in a domain and a single uniaxial domain wall with the average width of the experiment. After relaxation, the domain wall is formed with two imperfect David-star clusters, each with 12 Ta atoms, and a link atom between them (Fig. 2a). The simulated STM image of this optimized DW-1 structure is in reasonable agreement with the experimentally observed image of Fig. 2c showing three bright protrusions of the inverted triangular shape in the domain region and the regular triangles of less bright protrusions in the domain-wall region. The energy cost for the formation of a domain wall is as small as 0.03 eV per David star indicating that the phase transition can occur easily from the C-CDW to the NC-CDW phase at room temperature. The present DW-1 structure differs distinctly from those within the C-CDW phase at low temperature, so-called DW-2 or DW-4 structures,

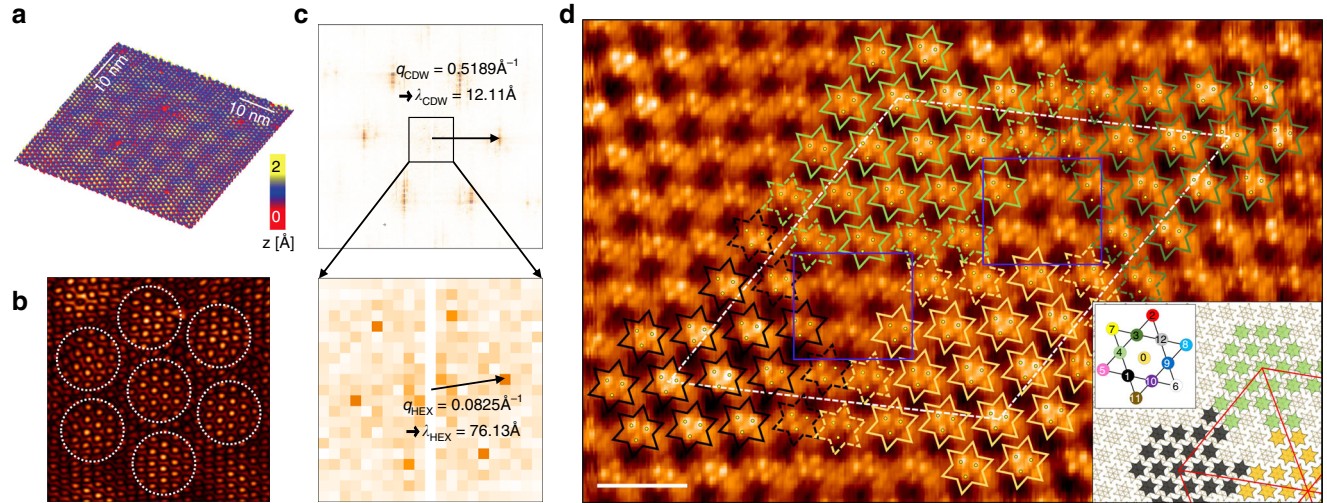

**Fig. 1** Nearly commensurate charge density wave phase of 1T-TaS$_2$ at room temperature. **a** Constant current STM image (sample bias voltage $V_s$ = −10 meV, tunneling current $I_t$ = 2 nA, and size $L^2$ = 50 × 50 nm$^2$). **b** Low-pass filtered image. White circles denote the hexagonal array of domain. **c** Fast Fourier transform of STM image. **d** Atomic resolution STM image ($V_s$ = −10 meV, $I_t$ = 3 nA). The solid and dashed David star denote the C-CDW clusters in four different domains and domain boundary, respectively. Filled (yellow) circles denote the top three S atoms in the center of David stars. Blue squares denote the area of vortex centers. *Scale bar*, 2 nm. Inset is a schematic arrangement of David stars for the NC-CDW phase

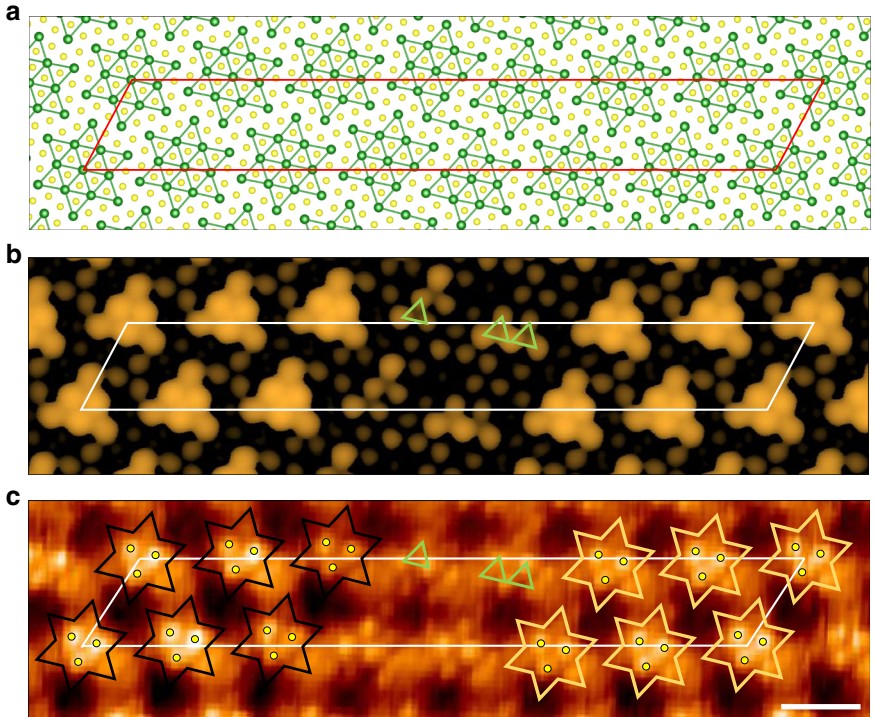

**Fig. 2** Atomic structure and STM images of DW-1. **a** Atomic structure, **b** simulated STM image ($V$ = −50 meV, $\rho$ = 1 × 10$^{-5}$ e Å$^{-3}$, and **c** experimental STM image. The green triangles denote the triangular shape of protrusions in domain-wall area. *Scale bar*, 1 nm

which have different displacements between domains and non-metallic density of states (DOS)[40]. These domain-wall configurations at low temperature have a less energy cost by 0.01 and 0.02 eV, which seems consistent with the thermally excited nature of NC-CDW domain walls.

**Confined metallic states within domain wall**. The electronic structure of the domain wall is significantly modified from the Mott-insulator state of the C-CDW phase. Figure 3a shows the

total DOS of the C-CDW phase with a Mott gap and the lower and upper Hubbard state at −0.12 and +0.27 eV, respectively. In clear contrast, the DW-1 structure exhibits nonzero DOS within the Mott gap indicating the metallic property with a shift of Hubbard states by 0.06 eV. As shown in more detail in Fig. 3b, the local DOS at a domain center and edges maintain the insulating character with a consistent Mott gap as the C-CDW phase. Thus, the in-gap or metallic states are strongly confined within the domain wall, that is, on the imperfect David stars (Fig. 3c).

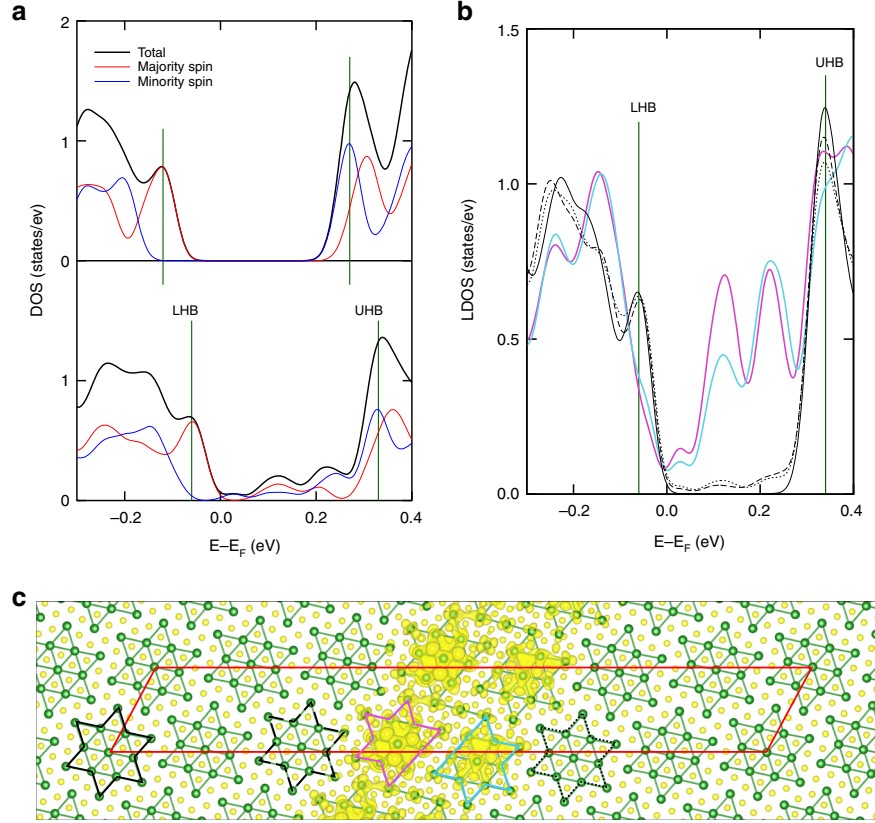

**Fig. 3** Density of states and charge character of DW-1. **a** Normalized total DOS of the C-CDW phase (upper panel) as a reference and DW-1 structure (lower panel). Vertical green lines denote the upper and the lower Hubbard band (UHB and LHB). **b** Local DOS at all Ta atoms in each David star of DW-1 structure. Each David star is denoted by solid and dashed line (domain center and edges) and colors (domain wall) in **c**. **c** Charge characters in the energy range from 0 to 0.2 eV

**Honeycomb network of domain walls.** We further construct the hexagonal domain lattice beyond a single DW-1 domain wall in order to describe the NC-CDW phase more closely as shown in Fig. 4. This model has a reduced (roughly 1/3) size of the domain with about seven David-star clusters for the calculation feasibility and neighboring domains are connected by the DW-1 structure. The total DOS of this model (inset of Fig. 4c) is consistent with the uniaxial DW-1 structure but much larger DOS appear in the Mott gap indicating a stronger metallic character. The extra in-gap states are due to the additional imperfect David stars in the domain-wall crossings or vortices mentioned above. As in the case of the uniaxial domain structure, the domain regions maintain the Mott-insulating state and the metallic states are well confined to the domain walls and vortices (see Fig. 4c and Supplementary Fig. 3). We thus can unambiguously conclude that the NC-CDW phase is metallic and the metallic states are localized in imperfect David-star clusters of the domain-wall network.

**Vortex and antivortex of domain walls.** As widely known, domain walls are topological excitations based on the degeneracy of the CDW ground state[45]. In the present honeycomb network, three neighboring DW-1 domain walls meet at an approximate $Z_3$ vortex or antivortex[45]. A vortex (antivortex) can be defined to have a clockwise (anticlockwise) rotation of phase-shift displacement vectors between involved domains (Fig. 4a). The four domains connected by a vortex–antivortex pair should have a proper relationship of their phase shift vectors, or connected topologically, for example, when a vortex connect 0-1-4 domains, the antivortex should connect 0-3-4 domains. After the full relaxation, a vortex and an antivortex have the same Ta atomic

structure (only rotated by 180°) shown schematically in Fig. 4b, c, making them electronically identical. Still, they have distinct structures in the S plane; one has a S atom at the center but the other does not. The structural difference of neighboring vortex and antivortex cores in the S plane is apparent in the STM images and their simulations (Fig. 4d). The charge analysis near the Fermi level (Fig. 4c) shows that the vortex (antivortex) has strong metallic electron density, which explain the extra contribution of metallic electrons in the NC-CDW phase. The nontrivial structural property of the vortex and antivortex pairs are due basically to the large structural degrees of freedom within them with many atoms to relax. To the best of our knowledge, this is the first case that the detailed vortex core structure of domain walls among various spin, dipole, and charge systems is disclosed.

## Discussion

The novel metallic domain wall network with vortex charges in the NC-CDW phase, as revealed here, can be a key to the phase transition from Mott state to superconductivity in 1T-TaS$_2$ under pressure and doping[13,15] and is consistent with the debated suggestion that the superconductivity forms within the metallic interdomain regions of the NC-CDW phase. We can also argue that the current picture, Mott-insulating domains with a metallic domain wall network, is valid under pressure and under the interlayer interaction. Our calculation indicates that the Mott state of C-CDW phase is robust against the in-plane (out-of-plane) lattice compression up to about 5 (6.7)%. This means that the Mott state of the domain regions of NC-CDW phase would also be robust. On the other hand, the interlayer interaction would affect the electronic structure of the NC-CDW phase by

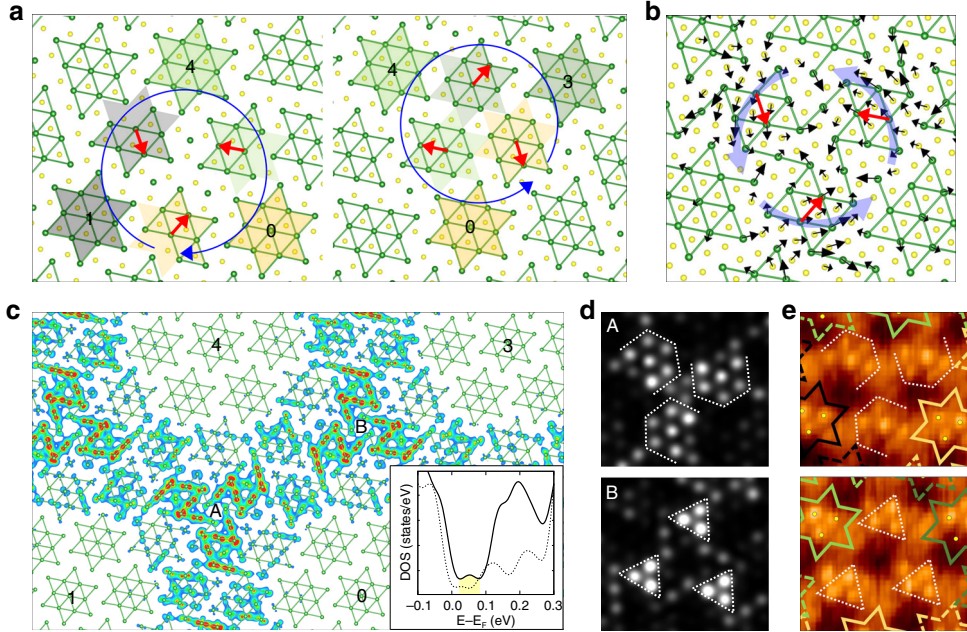

**Fig. 4** Atomic and electronic structure of domain-wall network. **a** Concept of the $Z_3$ vortex (clockwise, 0-1-4) and antivortex (anticlockwise, 0-3-4). Red arrows denote the phase-shift vectors between involved domains. The bottom S atoms are eliminated for better comparison of the top S atomic position at the vortex cores. **b** Optimized vortex core structure with atomic displacement (×15). Atomic displacement is obtained from the initial state of the clockwise case in **a**. **c** The vortex-related charge characters at Ta plane. The inset is a total DOS of the hexagonal domain model. Dashed line represents the total DOS of DW-1 for comparison. The vortex-related peak is denoted by yellow area. **d** Simulated STM images at vortex centers. Simulated images are obtained from the partial charge density of vortex-related state at a constant height of 2 Å away from the top S atom. **e** Experimental images are taken from Fig. 1d. The dashed lines are guides for eye. The different core structures are unambiguously distinguished by the bright and dark contrast at the centers of these images

the out-of-plane band dispersion and the change of the in-plane electronic structure by a possible stacking order[46–49]. These recent suggestions indicate metallic or small-gap insulating states, which are not consistent with the large Mott gap observed and are fragile against electron correlation (see Supplementary Fig. 4). The metallization scenario based on the gradual disordering of the stacking order[49] is not compatible with the abrupt transition between NC- and C-CDW phases[13] and the x-ray data[46]. While further investigation on the interlayer interaction seems necessary, we note that there is plenty of evidence that the intralayer effect is dominant for the low-temperature physics. For example, the change of the interlayer stacking order by intrinsic domain walls in the C-CDW phase does not affect the band gap[40], the out-of-plane resistivity is 500 times larger than the in-plane resistivity[50], and the superconducting phase is insensitive to the pressure[13] and the reduced dimensionality in thin flakes[38].

We further emphasize that this emergent network of the domain walls and vortices has generic nature to play an essential role in many-body quantum phenomena of 1T-TaS$_2$, such as the superconducting state. In general, domain walls host 1D metallic modes, which mix at vortices. Following a recent work for bilayer graphene[51], we theoretically model the system as a regular array of 1D metals living on the honeycomb lattice, which is valid at the energy scale smaller than the CDW gap. In the ideal limit where electrons scatter only at the nodes, the system harbors a series of the nearly flat bands with a small band width $\sim O((2\pi v_F)/L)$ (here $L$ is the spatial size of the domain wall) (see Supplementary Note 3). The bands also feature Dirac points, flat bands, and quadratic band touchings to the flat bands as shown in Fig. 5. The former is due to the honeycomb symmetry and the latter two produce huge DOS, which would amplify any small instability of the system and enhance the tendency toward many-body

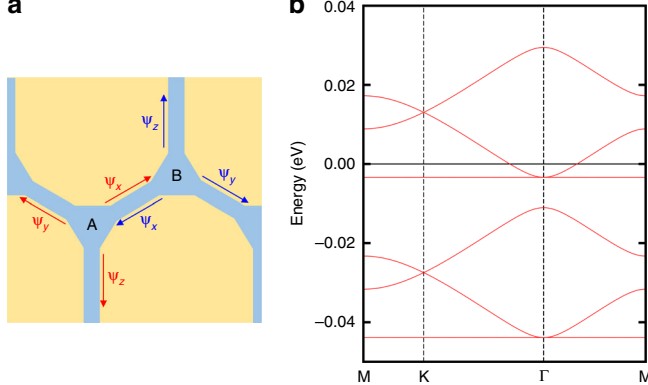

**Fig. 5** Honeycomb lattice model of domain-wall network. **a** Unit cell of network model. A and B denote the sublattice sites. $\psi_a$ and $\psi_{\bar{a}}$ represent the propagating direction of the modes. **b** Miniband structure ($v_F\hbar = 0.545$ eV Å and $L = 42.24$ Å). The value of $v_F\hbar$ is estimated by the band dispersion along the domain-wall direction of the DW-1 model. The same structures of minibands will repeat in energy to fill the entire Mott gap. Each one-dimensional bands inside the domain walls contribute a set of the minibands with different parameters, and hence we expect that many minibands will overlay each other in energy

correlated states including superconductivity. Such enhancement appears when the Fermi level is tuned near one of the cascade of the flat bands, and the tuning may be achieved by applying pressure or doping[13,15,36,37]. Moreover, the distinct atomic structure in the domain wall network can provide new low-energy phonon modes[52], which can favor the superconducting states as demonstrated in recent phenomenological theory[53]. Even

more interestingly, the strong correlation present in this system makes unconventional superconducting states expected theoretically[54–58]. The physics here is remarkably similar to that of twisted bilayer graphene with a small twist angle, which has a triangular network of 1D metals at the domain walls between locally BA-stacked and AB-stacked regions. Theoretically this system contains a set of nearly flat mini-bands of huge DOS[51,59], similar to those of our system, being crucial for the superconductivity and Mott phases[56–59]. Our work extends the bilayer graphene physics to various other 2D materials but suggests beyond it with the topological degrees of freedom provided by the honeycomb nature of the network and the inherently strong electron correlation. While the Dirac physics of the conducting honeycomb network and impact of disorders on the flat bands[60] call for further investigation, the controllability on domain wall network suggested in recent experiments would provide exciting possibilities for engineered quantum states in this class of materials.

## Methods

**STM measurement and sample preparations**. The STM measurements have been carried out with a commercial microscope (Omicron) at room-temperature. Pt–Ir wires were used as STM tips and topographic images were acquired in the constant current mode with bias voltage ($V_s$) applied to the sample. A single crystal of $1T-TaS_2$ was prepared by iodine vapor transport method. The samples were cleaved at room temperature and quickly transferred to the precooled STM head.

**DFT calculations**. DFT calculations were performed by using the Vienna ab initio simulation package[61] within the Perdew–Burke–Ernzerhof generalized gradient approximation[62] and the projector-augmented wave method[63]. The single-layer $1T-TaS_2$ was modeled with a vacuum spacing of about 13.6 Å. We used a plane-wave basis set of 259 eV and a $6 \times 6 \times 1$ $k$-point mesh for the $\sqrt{13} \times \sqrt{13}$ unit cell and atoms were relaxed until the residual force components were within 0.05 eV/Å. To more accurately represent electronic correlations, an on-site Coulomb energy was included for Ta $5d$ orbitals, which was estimated as 2.27 eV by the linear-response method[64]. This value reproduces the experimental Mott gap size[21]. To obtain the band spectrum of the honeycomb network model, we generalized the network model[51] on a triangular lattice to that on the honeycomb lattice. It is equivalent to the honeycomb array of scatterers of 1D metals and we solved it by using the standard Bloch wavefunction.

## Data availability

The authors declare that the data supporting the findings of this study are available within the article and its Supplementary Information.

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

## Acknowledgements

This work was supported by Institute for Basic Science (Grant no. IBS-R014-D1).

## Author contributions

H.W.Y. conceived the research idea and plan. J.W.P. performed the DFT calculations. G.Y.C. constructed the theoretical network model. J.L. performed the STM experiment. J.W.P., G.Y.C., and H.W.Y. prepared the manuscript with the comments of all other authors.

## Additional information

**Competing interests:** The authors declare no competing interests.

