## [Peer Review File · Nature Communications]

Reviewers' comments:

Reviewer #1 (Remarks to the Author):

J. Park et al. examined the well-known incommensurate charge density wave states called as a nearly commensurate charge density wave (NCCDW) state shown in 1T-TaS₂ compound using a scanning probe technique and computations. They claimed that the incommensurate regions are metallic and support a vortex like interpretation at the junctions of metallic regions. I found that the experiment observations are quite detailed and unprecedented. However, a title of the paper, phenomenological theories and interpretations seems to be overselling. So, some concrete evidences for the claims otherwise proper rewritings are suggested for further reviewing.

1) In the paper, the detailed atomic structures and related electronic properties along the hexagonal network incommensurate regions in the NCCDW are demonstrated using ab initio first-principles computations. However, I cannot find experiment spectroscopic data to compare themselves with experiment ones. Only the images are displayed. I suggest the authors present their spectroscopic data.

2) The title implies that the authors find a kind of topological excitation or that the authors understand the NCCDW states as topological excitations in 1T-TaS₂. Generally speaking, any incommensuration away from the commensurate states can support solitonic state. So, did authors imply this? If so, it would be better for the authors to explain some special things of this system other than the well known topological excitations in charge density wave systems. If not, please write the reason why they made the title.

3) The vortex like interpretation at the junction of three lines are interesting but I found no physical consequences from this interpretations. In Fig. 4d, the authors compare computations and experiment observations. This shows that the atomic displacements obtained in their first-principles computations agree well with the experiments and no more. In the supplement, quite interesting theoretical speculations are displayed but it seems that the theory is not relevant with 1T-TaS₂.

4) It has been known that the NCCDW state is accompanied with trilayer superlattice modulations as shown in Refs 11 and 27. The paper only deals with a first surface layer. Does the other layer or interlayer modulations affect the discussion? I think that the interesting theory in the supplement and discussion in the last part of the manuscript become to be irrelevant due to interlayer coupling.

4) 1T-TaS₂ becomes to be superconducting with electron doping following a suppression of NCCDW states (Refs. 13, 22, 38 and others). The transition temperature increases as metallic phase dominates. Moreover, the interlayer interactions are strong enough making the system be three-dimensional. Melting of Mott states through the metallic network in the NCCDW might be a possibility but no concrete evidences or clues are shown in the current manuscript. So, I suggest the authors modified the discussions seriously.

Reviewer #2 (Remarks to the Author):

Referee's comments on the manuscript NCOMMS-18-33392-T "Emergent Honeycomb Network of Topological Excitations in Correlated Charge Density Wave" by Park *et al.*

The authors present a study of the atomic and electronic structure of a particular domain wall network which forms within the so-called nearly commensurate (NC) charge density wave (CDW) phase in the prototypical layered CDW material 1T-TaS₂. They address the very important question whether or not these so-called discommensurations act as metallic channels and are, hence, responsible for the conducting transport properties of the NC-CDW phase. Based on supercell density functional theory (DFT) model calculations in comparison to scanning tunneling microscopy (STM) measurements they conclude that the discommensurations form a metallic honeycomb network. The authors further use a tight-binding model to study the properties of this metallic network and find striking similarities to the recently much discussed twisted bilayer graphene system.

In more detail, the authors use a fairly large 2D supercell which consists of several commensurate domains separated by discommensurations in order to model the domain wall structure of the NC-CDW phase. DFT+U was employed as pure LDA results in a metallic ground state for the commensurate structure without discommensurations. They then analyze the local density of states (LDOS) and the energy projected electron density to find that in-gap states form within the discommensurations network. Using this DFT model the authors simulate STM images and compare them to their experimental data.

Although these are very accurate and interesting calculations I think there are the following conceptual issues with the presented work:

(i) Most importantly, I think that one cannot neglect the interlayer correlations if one aims at a realistic description of this system. It has been shown experimentally by scanning probe measurements that these interlayer couplings are key for the transport properties of 1T-TaS₂ (see for instance Ref. [1]). Likewise, DFT indicates that the stacking arrangement of the CDW layers is crucial for the low energy electronic structure (Ref. [2, 3]). It would be highly interesting to extend the 2D model to three dimensions although I am aware that this might be computationally intractable.

(ii) Concerning the metallic properties of the discommensurations network: Some of the authors of the present work argued in their previous papers (Ref. [4, 5]) that the domain walls are clearly non-metallic. I am wondering if their new STM/STS data clearly indicate insulating domains and metallic discommensurations? Why is no dI/dU spectra shown? As far as I can see at least the first DFT model (Fig. 2 and 3) in the current paper was already presented in the previous paper (Ref. [5]). I do not really understand why the authors reach a different conclusion this time and argue in favor of metallic domain walls?

(iii) Also the question whether or not local electron-electron correlations are necessary in order to describe the commensurate CDW in 1T-TaS₂ is a matter of debate. Here the authors use DFT+U to obtain a ≈ 200 meV gap for their strictly 2D model which would be metallic in pure DFT. However, transport measurements and photoemission experiments never observe such a large gap. Rather these methods find gaps on the order of several meV (see for instance Ref. [6–8]) which raises the question whether the surface gap is much larger than the bulk gap and if so, why? Also, for DFT+U one needs to assume some sort of magnetic ordering. However, so far there has been no evidence for magnetic ordering in 1T-TaS₂.

(iv) Another point that puzzles me is that the authors claim that their calculations indicate that the Mott state would be robust against an in-plane compression of about 5%. Experiments however show that hydro static pressure as little as ≈ 1 GPa, which certainly does not yield a 5% compression, is sufficient to suppress the C-CDW phase in favor of the NC-CDW phase. I think that indicates once more that interlayer effects (the *c*-axis is much softer with respect to compression) are more important than the in-plane compression for the pressure induced C-NC transition.

In summary, the authors address a very interesting and important question concerning the electronic and atomic structure of networks of discommensurations. However, by considering effectively a monolayer of 1T-TaS₂ they completely neglect interlayer couplings which I think is not a valid approach to describe the low energy structure of this system. Also I am not sure about the degree of novelty of the present paper as at least parts of the theoretical model and experimental data were already presented in a previous paper, although with a different conclusion. Therefore I do not recommend publication of the manuscript in the current form in Nature Communications.

References

- [1] L. Ma et al., Nat Commun **7**, 10956 (2016) [10.1038/ncomms10956](https://doi.org/10.1038/ncomms10956).
- [2] Y. Ge et al., Phys. Rev. B **82**, 155133 (2010) [10.1103/PhysRevB.82.155133](https://doi.org/10.1103/PhysRevB.82.155133).
- [3] T. Ritschel et al., Nat Phys **11**, 328 (2015) [10.1038/nphys3267](https://doi.org/10.1038/nphys3267).
- [4] D. Cho et al., Nat Commun **7**, 10453 (2016) [10.1038/ncomms10453](https://doi.org/10.1038/ncomms10453).
- [5] D. Cho et al., Nat. Commun. **8** (2017) [10.1038/s41467-017-00438-2](https://doi.org/10.1038/s41467-017-00438-2).
- [6] D. Svetin et al., Scientific Reports **7**, 46048 (2017) [10.1038/srep46048](https://doi.org/10.1038/srep46048).
- [7] A. S. Ngankeu et al., Physical Review B **96** (2017) [10.1103/physrevb.96.195147](https://doi.org/10.1103/physrevb.96.195147).
- [8] M. Bovev et al., Phys. Rev. B. **69**, 125117 (2004) [10.1103/PhysRevB.69.125117](https://doi.org/10.1103/PhysRevB.69.125117).

Reviewer #3 (Remarks to the Author):

The authors present an interesting paper about the detailed atomic and electronic domain wall structure in the NC-CDW phase of 1T-TaS₂. Room temperature STM experiments are complemented by DFT calculations. The authors show that the domain walls are associated with particular types of imperfect David-star clusters within the CDW state, which are imaged by STM. The domain walls networks becomes connected by Z₃ topological vortices that form a honeycomb structure. The metallic states are confined within the domain walls. They suggest that this gives rise to symmetry protected Dirac bands and flat bands, which are suggested to be important for the emerging superconductivity in 1T-TaS₂. A general theoretical framework proposed by the authors gives predictions that are consistent with published experimental works on 1T-TaS₂ under pressure and doping.

There are a number of small corrections proposed for the paper:

Fig 2c) the major atomic protrusions in the domain wall - depicted as regular triangles are not very clearly visible, perhaps a higher resolution image can be included as an inset.

Fig 3c) what are the pink and blue lines in the DOS calculations?

The discommensuration engineering proposed by the authors is difficult to control in practice, perhaps this statement can be either clarified or softened.

Provided the following comments are addressed, the paper would be recommended for publication. It provides novel interesting results and addresses very topical and general concepts of flat band superconductivity and topological vortex states which are of great interest to the scientific community.

Replies to the comments of the first reviewer

Comment : J. Park et al. examined the well-known incommensurate charge density wave states called as a nearly commensurate charge density wave (NCCDW) state shown in 1T-TaS₂ compound using a scanning probe technique and computations. They claimed that the incommensurate regions are metallic and support a vortex like interpretation at the junctions of metallic regions. I found that the experiment observations are quite detailed and unprecedented. However, a title of the paper, phenomenological theories and interpretations seems to be overselling. So, some concrete evidences for the claims otherwise proper rewritings are suggested for further reviewing.

Reply) We appreciate the high evaluation of our work by the reviewer and his/her helpful comments. The reviewer addressed five important and constructive comments of which we provide detailed considerations as listed below.

Comment 1) In the paper, the detailed atomic structures and related electronic properties along the hexagonal network incommensurate regions in the NCCDW are demonstrated using ab initio first-principles computations. However, I cannot find experiment spectroscopic data to compare themselves with experiment ones. Only the images are displayed. I suggest the authors present their spectroscopic data.

Reply 1) We appreciate this important comment. Following this comment, we added our spectroscopy data in the revised supplementary material with a short discussion. However, unfortunately, the room temperature STS data do not provide crucial information due to the substantial thermal broadening of the spectral features. They tell only the global metallic nature of the sample with the difference between the domain and the domain wall regions wiped out. As shown below, this spectral broadening can roughly be expected from the simulation of thermal broadening of the density of states in the first principles calculation (which is so called the Gaussian smearing method considering only the electron temperature [Phys. Rev. B **82**, 155133 (2010) and Phys. Rev. B **88**, 121403 (2013)]). Low temperature STS measurements of the NCCDW phase can differentiate the domain wall metallic states but is possible only for the doped sample with the suppressed transition temperature into the commensurate CDW phase. These samples with reliably quality seem hard to fabricate and is not available at present.

Fig. 2 STS spectra of the NC phase and theoretical DOS of DW-1 structure. (a) STS spectra, (b) Total DOS as a function of the electronic temperature parameter σ (eV).

Comment 2) The title implies that the authors find a kind of topological excitation or that the authors understand the NCCDW states as topological excitations in 1T-TaS₂. Generally speaking, any incommensuration away from the commensurate states can support solitonic state. So, did authors imply this? If so, it would be better for the authors to explain some special things of this system other than the well known topological excitations in charge density wave systems. If not, please write the reason why they made the title.

Reply 2) We note this comment is based on the limited understanding of our major points. We acknowledge that our meaning of the ‘topological excitation’ is based on a solitonic state as the referee indicated. This is generally true for domain walls in incommensurate CDW phases. However, the word ‘topological excitations’ here can be simply replaced by ‘domain walls,’ since the present work goes beyond the simple soliton concept in two points. First, we identified an extra type of topological excitations, the vortices of domain walls. The domain wall vortices are natural consequences of a domain wall network in 2D systems as well documented recently [Ref. 46]. The ‘vortex’ concept itself [Ref. 46] is natural due to the rotating atomic (and CDW) displacement vectors with two different chirality in domain wall joints (Fig. 4). To the best of our own knowledge, the present report is the first to reveal the detailed atomic structures and the distinct electronic state of domain wall vortices. Second, in a multiply degenerate system like 1T-TaS₂ CDW with 13 Ta atoms within a unitcell, there can be twelve different domain wall configurations [Refs. 33 and 40]. This provides various different electronic states for domain walls as illustrated recently [Sci. Adv. **4**, eaau5501 (2018)]. The electronic states of domain walls in the present system can be metallic or insulating depending on odd or even atoms within domain wall unitcells. This diversity cannot be explained by the simple soliton concept. We thus believe that ‘topological excitations’ is a proper terminology to describe two distinct topological defects of domain walls and their joints (vortices) and the diverse electronic states of domain walls. However, if the reviewer insists on this issue, we may replace ‘topological excitations’ by ‘domain wall and their vortices.’

Comment 3) The vortex like interpretation at the junction of three lines are interesting but I found no physical consequences from this interpretations. In Fig. 4d, the authors compare computations and experiment

observations. This shows that the atomic displacements obtained in their first-principles computations agree well with the experiments and no more. In the supplement, quite interesting theoretical speculations are displayed but it seems that the theory is not relevant with 1T-TaS₂.

Reply 3) This is closely related to the previous comment and we largely disagree with these comments. The vortex interpretation of domain wall junctions is rather well established one [Ref. 46] and has important physical consequences. First of all, as fully discussed in the manuscript, the characteristic atomic structures of domain vortices induce distinct electronic states from domain walls, providing extra metallic electrons to the domain wall network. Higher metallic density of states would in principle be important for the emerging superconductivity. This is also essential for our network theory since they decide the scattering processes of domain wall conducting electrons and thus the low-energy band structures. We strongly disagree with the referee's argument that our network theory is not relevant with 1T-TaS₂. This theory is a generic one to calculate the emergent band structure of a honeycomb network of conducting electrons and we showed that the NCCDW phase of 1T-TaS₂ indeed features such a conducting honeycomb network. Furthermore, this band structure provides singular contribution to density of states, which is inarguably crucial for understanding one of the most important problems of this material, emerging superconductivity. Thus, this model is obviously relevant to the present system. If one goes into further details of this model, it is important that two neighboring vortices have the same electronic states to guarantee the same scattering. This is not trivial with the different atomic displacements of neighboring vortices as shown in Fig. 4. However, we clarified that two neighboring vortices (or a vortex and an anti-vortex) are electronically identical. Thus, the detailed study of the vortices and the generic conducting network theory are well correlated.

Comment 4) It has been known that the NCCDW state is accompanied with trilayer superlattice modulations as shown in Refs 11 and 27. The paper only deals with a first surface layer. Does the other layer or interlayer modulations affect the discussion? I think that the interesting theory in the supplement and discussion in the last part of the manuscript become to be irrelevant due to interlayer coupling.

Reply 4) We appreciate the reviewer for pointing this important issue of the interlayer coupling out. We acknowledge that this issue was not thoroughly discussed in our original manuscript. As the reviewer mentioned, the interlayer coupling effect was suggested to be important in 1T-TaS₂ in a few recently papers [Nat. Phys. 11, 328 (2015) and PRB 98, 195134 (2018)]. However, a large majority of the literature still discuss the present system within the 2D single-layer physics [Refs. 23, 24, 32, 33, 36, 40] and the 2D Mott-CDW physics [Refs. 13, 17-24, 33-38, 40, 41, 44], as we did here. Thus, one has to admit at least that currently there is no clear consensus on how crucial the interlayer coupling is. Below and in the revised manuscript, we provide more extensive arguments that the interlayer coupling seems not too crucial for our discussion and conclusions.

- (i) There exists a large volume of experimental evidence that the system is governed by the intralayer 2D physics. For an important example, the resistivity of out-of-plane is 500 times larger than the in-plane resistivity even in the metallic phase [Physica B+C 99, 173 (1980)]. Indirectly speaking, the interlayer interaction should be much stronger under high hydrostatic pressure, but the superconductivity enhances and is maintained under high pressure [Ref. 13]. More directly, the superconducting phase remains intact from the reduced dimensionality for thin flakes of 1T-TaS₂ [Ref. 38]. Therefore, while the finite interlayer coupling (the finite out-of-plane dispersion) is present, it is thought to be a secondary effect in the metallization and the superconductivity.
- (ii) While the recent ARPES study showed a finite out-of-plane dispersion, it is not conclusive on the metallic behavior. This study showed a band dispersing up to the Fermi level but ARPES cannot detect the DOS above the Fermi energy to conclude a metallic dispersion unambiguously. In contrast, STS, detecting both filled and empty DOS, showed that the Fermi level is very close to the top of conduction band in most of the samples due to unintentional doping with a large band gap above the Fermi energy [Refs. 20, 33, and 34]. This is apparently consistent with the ARPES result. Moreover, a few other ARPES studies [Refs. 38 and Nat. Comms, **3**, 1069 (2012)] are consistent with the Mott-CDW gap. Furthermore, laser-excited ARPES can probe the spectral feature above the Fermi energy and concluded the insulating state at low temperature, which becomes metallic in the excited state [New J. Phys. **10**, 053019 (2018) and PRL **120**, 166401 (2018)].
- (iii) As the referee indicated, a tri-layer stacking order was reported in the NC phase [Structural Dynamics 4, 044020 (2017)]. The experimental data have not been confirmed yet and a few other reports denied the existence of any ordered stacking structure [PNAS **1113**, 11420 (2016)]. In first-principle calculations without electron-electron correlations, the interlayer stacking was reported to induce the in-plane metallization [Nat. Phys. **11**, 328 (2015) and PRB **98**, 195134 (2018)]. However, very obviously, this metallic band structure cannot explain the large band gap of CCDW phase and the major spectral features around the Fermi level confirmed in a number of STS works [Refs 20, 33, 34]. As shown clearly in our own calculations (figure below), such a tri-layer stacking order gives a huge metallic density of states in clear distinction from experimental data. Including the electron correlation readily destroys the metallic phase induced by the tri-layer interlayer coupling as calculated here (figure below). Thus, we can unambiguously conclude that the commensurate domain region remains in the Mott insulating phase even with the interlayer coupling, making the metallization of domain walls becomes crucial.

Ref. 20 STS spectra of the Mott insulating CCDW phase.

SFig. 4 One possible tri-layer stacking order in the C phase. (a) atomic structure, (b) and (c) are the band structure and total DOS without and with electron-electron correlations, respectively. The interlayer distance of 5.9 Å is fixed.

- (iv) Then, what is left in our argument is whether the domain wall electronic states are affected by the interlayer coupling or not. When the domains remain insulating, we generically expect the effect of the interlayer couplings to be weak for the conducting domain wall network. The reason is that the leading effect of the interlayer coupling is the electron hopping between the layers, which are strongly suppressed if not the networks of neighboring layers are aligned to be almost overlapping. Within the currently available experimental data, e.g., the data from [Ref. 33, and 40], the networks in different layers are not correlated to those of the other layers (the second layer domain walls are barely noticed in the left figure below, taken from [Ref. 33]). We can further show that even the interlayer coupling of domain walls cannot destroy the metallic property of a domain wall. See the calculations given in the figure below for two different stackings. Hence, the interlayer coupling can safely be excluded for any substantial effect on the low-energy electronic modes, which mainly originate from conducting domain walls and their many-body physics.

Motivated from the reviewer's comment, we added some of the discussion to the supplemental materials.

Ref. 33 Existence of the subdomains and the stacking order. The black arrows indicate the domain walls in the sublayers.

SFig. 5 Two different bilayer stacking order in the DW-1 structure. (a) on-top stacking and (b) shifted stacking. Top and bottom panels are atomic structure and LDOS, respectively. All atoms are fixed at their atomic position in the single-layer DW-1 structure. The interlayer distance of 5.9 \AA is fixed.

Comment 5) 1T-TaS₂ becomes to be superconducting with electron doping following a suppression of NCCDW states (Refs. 13, 22, 38 and others). The transition temperature increases as metallic phase dominates. Moreover, the interlayer interactions are strong enough making the system be three-dimensional. Melting of Mott states through the metallic network in the NCCDW might be a possibility but no concrete evidences or clues are shown in the current manuscript. So, I suggest the authors modified the discussions seriously.

Reply 5) This is not a valid comment. The reviewer seems to confuse the CCDW and the NCCDW phase. The superconductivity follows the suppression of CCDW but not NCCDW, while NCCDW T_c also decreases. It becomes obvious from the fact that the superconducting T_c reaches to its maximum of $\sim 4 \text{ K}$ when T_c of NCCDW becomes as large as $\sim 200 \text{ K}$, two orders of magnitude difference. The superconducting phase indeed locates not outside but well 'within' the NCCDW phase as shown in the well established phase

diagrams of Refs. 22 and 38. It should also be noted that most of the current discussion on the origin of the emerging superconductivity relates the domain wall network of NCCDW with the superconductivity [Refs. 13, 14]. While the experimental evidence of the relationship between the NCCDW phase and the superconductivity is affluent, what has been lacking is the mechanism to connect them. In the present work, we showed that the domain wall network is a conducting honeycomb network and it can generically induce superconductivity.

Ref.13

Ref. 22

Ref. 38

Replies to the comments of the second reviewer

Comment: The authors present a study of the atomic and electronic structure of a particular domain wall network which forms within the so-called nearly commensurate (NC) charge density wave (CDW) phase in the prototypical layered CDW material 1T-TaS₂. They address the very important question whether or not these so-called discommensurations act as metallic channels and are, hence, responsible for the conducting transport properties of the NC-CDW phase. Based on supercell density functional theory (DFT) model calculations in comparison to scanning tunneling microscopy (STM) measurements they conclude that the

discommensurations form a metallic honeycomb network. The authors further use a tight-binding model to study the properties of this metallic network and find striking similarities to the recently much discussed twisted bilayer graphene system. In more detail, the authors use a fairly large 2D supercell which consists of several commensurate domains separated by discommensurations in order to model the domain wall structure of the NC-CDW phase. DFT+U was employed as pure LDA results in a metallic ground state for the commensurate structure without discommensurations. They then analyze the local density of states (LDOS) and the energy projected electron density to find that in-gap states form within the discommensurations network. Using this DFT model the authors simulate STM images and compare them to their experimental data. Although these are very accurate and interesting calculations I think there are the following conceptual issues with the presented work.

Reply) We appreciate the high evaluation of our work by the reviewer and his/her helpful comments. The reviewer addressed four important and constructive comments of which we provide detailed considerations as listed below.

Comment 1) Most importantly, I think that one cannot neglect the interlayer correlations if one aims at a realistic description of this system. It has been shown experimentally by scanning probe measurements that these interlayer couplings are key for the transport properties of 1T-TaS₂ (see for instance Ref. [1]). Likewise, DFT indicates that the stacking arrangement of the CDW layers is crucial for the low energy electronic structure (Ref. [2, 3]). It would be highly interesting to extend the 2D model to three dimensions although I am aware that this might be computationally intractable.

Reply 1) We appreciate the reviewer for pointing this important issue out. This issue was also shared by the first reviewer. We acknowledge that this issue was not thoroughly discussed in our original manuscript. We provide extensive arguments and extra calculations that the interlayer coupling is not crucial to the present discussion and conclusions for the comment 4 of the reviewer 1 (please see above). In short, there is no clear consensus on existence of the ordered interlayer coupling and the crucial role of it in the present system. On the other hand, we can clearly show with our own first-principle calculations that the Mott gap of the domain region and the metallic states at the domain walls are preserved even under the interlayer coupling. (We have added extra discussion about the effect of interlayer coupling on the conducting network model in the supplemental material.)

Comment 2) Concerning the metallic properties of the discommensuration network: Some of the authors of the present work argued in their previous papers (Ref. [4, 5]) that the domain walls are clearly non-metallic. I am wondering if their new STM/STS data clearly indicate insulating domains and metallic

discommensurations? Why is no $dI=dU$ spectra shown? As far as I can see at least the first DFT model (Fig. 2 and 3) in the current paper was already presented in the previous paper (Ref. [5]). I do not really understand why the authors reach a different conclusion this time and argue in favor of metallic domain walls?

Reply 2) We appreciate the reviewer for pointing this issue explicitly but note that there exists important misunderstanding here. The current domain wall structure is absolutely different from what we reported in the previous work [Ref. 5] on the low temperature CCDW phase. Among 12 different DW structures possible (as classified in [Refs. 33 and 40]), the low temperature CCDW phase has DW-2 and DW-4 structures, which have two and four Ta atoms per $\sqrt{13}$ period and are non-metallic. The current domain wall is DW-1 structure with only one Ta atom missing and is metallic as shown here. That is, the major finding of the present work is that the CCDW phase and the NCCDW phase have totally different domain wall structures, which explains the unique metallic phase of NCCDW.

As mentioned for the comment 1 of the first reviewer, unfortunately, the room temperature STS data (added above and in the supplementary information) do not provide crucial information due to the substantial thermal broadening of the spectral features but only tell the global metallic nature of the sample. The difference between the domain region and the domain wall is wiped out, too. Low temperature STS measurements of the NCCDW phase can differentiate this but is possible only for the doped sample with the suppressed transition temperature into the commensurate CDW phase. These samples with reliably quality is not available at present.

Comment 3) Also the question whether or not local electron-electron correlations are necessary in order to describe the commensurate CDW in 1T-TaS₂ is a matter of debate. Here the authors use DFT+U to obtain a 200 meV gap for their strictly 2D model which would be metallic in pure DFT. However, transport measurements and photoemission experiments never observe such a large gap. Rather these methods find gaps on the order of several meV (see for instance Ref. [6-8]) which raises the question whether the surface gap is much larger than the bulk gap and if so, why? Also, for DFT+U one needs to assume some sort of magnetic ordering. However, so far there has been no evidence for magnetic ordering in 1T-TaS₂.

Reply 3) We appreciate the reviewer for pointing this important issue out. This is closely related to the interlayer coupling issue which is detailed above for the comment 4 of the first reviewer. We add some more comments here. As discussed in detail above, ARPES does not in principle tell the band gap but only the position of the valence band top. As the reviewer indicated this top is very close to the Fermi level and is consistent with most of the STS data, which showed clear gap above the Fermi energy. That indicates that most of the samples are insulating and electron-doped. More detailed discussion of the spectroscopy data is given above. In the case of the transport measurement [reviewer's Ref. 6], the very small action energy (~ 91 K

or ~ 8 meV) was reported in the temperature range of 40-140K. However, this work attributed the data not to the metallic band structure but to the transport barriers for polarons.

As also discussed above, the interlayer coupling can induce metallic state but is not consistent with the STS data. One should include the electron correlation to explain the spectroscopic gap [Refs 13, 17-24, 33-38, 40, 41, 44]. That is, the Mott physics in 1T-TaS₂ is widely accepted in both theory and experiments. Of course, as mentioned by the reviewer, including U correction in DFT predicts a magnetic ordering and it looks inconsistent with no apparent magnetic ordering in the 1T-TaS₂. However, a recent experiment (specific heat, magnetization, neutron diffraction, and μ SR measurements) supported the magnetic ordering of 1T-TaS₂ with $S = 1/2$ moments localized in the centers of the David stars [NPJ Quantum Materials 2, 42 (2017)] at very low temperature. At higher temperature, the strong spin frustration was suggested for the present system [Refs 24 and 25]. Therefore, the Mott physics of the current system seems still valid with a strong spin frustration generic to a triangular lattice.

Comment 4) Another point that puzzles me is that the authors claim that their calculations indicate that the Mott state would be robust against an in-plane compression of about 5%. Experiments however show that hydro static pressure as little as 1 GPa, which certainly does not yield a 5% compression, is sufficient to suppress the C-CDW phase in favor of the NC-CDW phase. I think that indicates once more that interlayer effects (the c-axis is much softer with respect to compression) are more important than the in-plane compression for the pressure induced C-NC transition.

Reply 4) We appreciate the referee for pointing this rather unclear part out. We agree with the reviewer on the fact that the CCDW phase is suppressed by small hydrostatic pressure. This indicates that the pressure generates domain walls for the NCDW phase. What we claimed is that not the CCDW phase but the Mott phase (or Mott gap) is consistent under such pressurization; the NCCDW phase is composed of domains with a Mott gap and the conducting domain walls. That is, we argue that the Mott physics of the domain regions are intact, which is the backbone of our claim for the emerging superconductivity. In order to avoid any further misunderstanding, the corresponding part of the manuscript was revised.

Comment In summary, the authors address a very interesting and important question concerning the electronic and atomic structure of networks of discommensurations. However, by considering effectively a monolayer of 1T-TaS₂ they completely neglect interlayer couplings which I think is not a valid approach to describe the low energy structure of this system. Also I am not sure about the degree of novelty of the present paper as at least parts of the theoretical model and experimental data were already presented in a previous paper, although with a different conclusion. Therefore I do not recommend publication of the manuscript in the current form in Nature Communications.

Reply) We disagree with these comments. As detailed above, the interlayer coupling issue can be shown to be not crucial with vast data accumulated in literature and with our own calculations. In clear contrast to the reviewer comment, our data and structure model are completely original and have never been exposed before in the previous work. In particular, the precise role of domain walls and vortices in connection of large density of states and emerging superconductivity that we uncover in this paper is entirely new. This was explicitly acknowledged by two other reviewers.

Replies to the comments of the third reviewer

The authors present an interesting paper about the detailed atomic and electronic domain wall structure in the NC-CDW phase of 1T-TaS₂. Room temperature STM experiments are complemented by DFT calculations. The authors show that the domain walls are associated with particular types of imperfect David-star clusters within the CDW state, which are imaged by STM. The domain walls networks becomes connected by Z₃ topological vortices that form a honeycomb structure. The metallic states are confined within the domain walls. They suggest that this gives rise to symmetry protected Dirac bands and flat bands, which are suggested to be important for the emerging superconductivity in 1T-TaS₂. A general theoretical framework proposed by the authors gives predictions that are consistent with published experimental works on 1T-TaS₂ under pressure and doping.

Comment 1) Fig 2c) the major atomic protrusions in the domain wall - depicted as regular triangles are not very clearly visible, perhaps a higher resolution image can be included as an inset.

Reply 1) The reviewer suggested that a higher resolution image of Fig. 2(c) can be included as an inset. Unfortunately, due to the thermal broadening, which is serious in the present room temperature measurement, the Fig. 2(c) is the best image we have. Instead, we provide a more well filtered image in the supplementary information.

Comment 2) Fig 3c) what are the pink and blue lines in the DOS calculations?

Reply 2) The reviewer asked about the pink and blue lines in the DOS calculations. Following this advice, we provide more clear information in the caption of Fig. 3(b).

Comment 3) The discommensuration engineering proposed by the authors is difficult to control in practice, perhaps this statement can be either clarified or softened. Provided the following comments are addressed, the

paper would be recommended for publication. It provides novel interesting results and addresses very topical and general concepts of flat band superconductivity and topological vortex states which are of great interest to the scientific community.

Reply) Following this advice, we rephrase the claim of the discommensuration engineering.

3. Summary of changes:

(1) We added four figures in Supplementary Materials.

- SFig. 1 Well filtered STM image of the NC phase.
- SFig. 2 STS spectra of the NC phase and theoretical DOS of DW-1 structure.
- SFig. 4 Tri-layer stacking order in the C phase.
- SFig. 5 Bilayer stacking order in the DW-1 structure.

(2) We changed sentences in result to discuss the inter-layer correlations.

Our calculation further indicates that the Mott state of C-CDW phase is robust against the in-plane lattice compression up to about 5% and the interlayer interaction would be a secondary effect to the in-plane conductivity. Note also that the transition temperature is rather insensitive to the pressure within the superconducting regime [13]. The above considerations make the role of network more persuasive for the superconductivity emerging in the high pressure.

=>

We can also argue that the current picture, Mott-insulating domains with a metallic domain wall network, is valid under pressure and under the interlayer interaction. Our calculation further indicates that the Mott state of C-CDW phase is robust against the in-plane lattice compression up to about 5%. This means that the Mott state of the domain regions of NC phase would also be robust. On the other hand, the interlayer interaction would affect the electronic structure of the NC phase by the out-of-plane band dispersion and the change of the in-plane electronic structure by a possible stacking order [Nat. Phys. 11, 328 (2015), PRB 98, 195134 (2018), STRUCTURAL DYNAMICS 4, 044020 (2017)]. These works indicate metallic or small-gap insulating states, which are not consistent with the large Mott gap observed and fragile against electron correlation. Note also that there is plenty of evidence that the intralayer physics is dominant for the low temperature physics. The out-of-plane resistivity is 500 times larger than the in-plane resistivity [Physica B+C 99, 173 (1980)] with the superconducting phase insensitive to the pressure [Ref 13] and the reduced dimensionality in thin flakes [ref. 38].

(3) We changed a sentence in caption of Fig. 3(b).

(b) Local DOS in each David star of DW-1 structure and (c) Charge characters in the energy range from 0 to 0.2 eV. The color and dashed lines in (b) correspond to the each David star in (c).

=>

(b) Local DOS at all Ta atoms in each David star of DW-1 structure. Each David star is denoted by solid and dashed lines (domain center and edges) and colors (domain wall) in (c), and (c) Charge characters in the energy range from 0 to 0.2 eV.

(4) We added four papers in reference.

- Physica B+C 99, 173 (1980)
- Nat. Phys. 11, 328 (2015)
- STRUCTURAL DYNAMICS 4, 044020 (2017)
- PRB 98, 195134 (2018)

(5) We have corrected a typo in the main text:

“...which has a triangular network of one-dimensional metals at the domain walls between locally ~~AA-~~ BA- and AB- stacked regions...”

(6) We have added some discussion in the supplementary materials about the effect of interlayer coupling in the conducting honeycomb network of domain walls:

In this subsection, we consider the effect of interlayer coupling to the electronic structures in the conducting network, and we will argue that, in general, the interlayer couplings between the layers will little affect the emergent electronic structures.

For the concrete-ness of our theoretical discussion, we first assume that the charge-density wave domains in the nearly commensurate phase remain insulating even after the inclusions of interlayer couplings. With this in hand, all the lowest-energy electronic states are in the domain walls in the conducting networks, and the interlayer couplings will introduce the coupling between these metallic modes inside the conducting networks living in different layers.

Among various possible couplings, the most important coupling, which can largely modify the band structure, is the electron hopping process between the layers. Note that this is proportional to the wavefunction overlap between the states of domain walls in different layers, and the states are highly localized within each domain walls. Hence, the effect of coupling will be strongly suppressed if not the networks are almost exactly overlapping to each other when seen from c-axis. From the available literature, we note that the networks in different layers are not correlated to each other and thus we expect that the emergent band structure of the low-energy theory will not be affected much by the interlayer coupling.

Reviewers' comments:

Reviewer #1 (Remarks to the Author):

The authors of the manuscript answered questions and comments thoroughly. I found that all answers and modifications according to referee reports are more or less satisfactory. However, in some answers and in some revised sentences, there may be some issues to be discussed further.

1) In their reply 3, the authors argue that the topological networks or vortex structures are important because they are responsible for superconductivity. However, as shown in Ref. 13 and Ref. 38, the sample becomes a superconductor only with pressure and doping. With pressure, owing to its layered nature (large difference in in-plane and out-of-plane Young's modulus), TaS₂ should shrink a lot in direction perpendicular to the layer not in parallel direction. So, suitable theoretical consideration may use reduced interlayer distance or highly doped sample. For this situation, I wonder how this hexagonal network contribute to superconductivity with those conditions. It should be clearly written reason why the existing hexagonal metallic network in a pristine NCCDW sample contribute superconductivity only after pressure applied (Ref. 13) or high ionic gate doping (Ref. 38).

Moreover, in Fig. 5(b), the bands of hexagonal network shows flat bands or very narrow bands width of few meV. The authors argue that those mini bands can fill the all the Mott gap and can contribute the superconductivity. However, considering disorders (distortion of hexagons etc) shown in NCCDW phase in Fig. 1a, the large scale hexagonal network seems not so much clean compared with twisted bilayer graphene and their small group velocity of energy bands may be altered a lot with disorders.

As I wrote in my previous report, atomic vortex structures are interesting observations but their connections to other physics in TaS₂ are not directly supported by authors' experiment observations. Interesting band structures of hexagonal vortex network may be relevant with some observables in this compound also. But this also again lacks direct connections with superconductivity in doped or pressurized 1T-TaS₂.

Reviewer #2 (Remarks to the Author):

The authors kindly answered all questions raised in my previous report. More precisely, they addressed the following points:

(i): As also noted by the first Referee the original version of the manuscript did not discuss the role of interlayer coupling. The authors largely expanded this discussion in the revised version and also provide additional calculations and conclude that interlayer correlations are not relevant for the main result of their paper. Even though, I still think that interlayer coupling must play a significant role for the CCDW-NCCDW transition - simply because the stacking order of the CDW changes drastically from a disordered stacking in the CCDW to a well-ordered stacking in the NCCDW - I agree with the authors that, within their model, interlayer coupling is secondary for the metallization. However, the experimental verification of the DW metallization is still inconclusively (see also point (ii)).

(ii): I thank the authors for clarifying to me that the DW type 1 studied in the current paper is indeed distinct from the previously investigated DWs 2 and 4 - a very important piece of information that must have slipped my attention in the original manuscript.

I also appreciate the addition of STS data for the NC phase. Unfortunately, these data do not show much contrast between DW, domain and vortex. The authors attribute the absence of contrast to

unavoidable thermal broadening at room temperature and show theoretical DOS with different Gaussian smearing. For values of the smearing parameter $\sim > 0.1$ eV the theoretical spectra show a smeared out gap similar to the experimental data. However, 0.1 eV corresponds to about 1000K which is rather high compared to 300K.

Also, I would like to see the theoretical spatial resolved LDOS for DW, domain and vortex with different smearing parameters in order to verify that the contrast between domain, DW and vortex indeed gets wiped out.

(iii): I appreciate the expanded discussion about the interlayer coupling in the revised manuscript which gives a more balanced view of the debate whether or not strong electron-electron correlations are important for the low temperature phase of 1T-TaS₂. Nevertheless, I suppose that it strongly depends on the specific stacking and the imposed magnetic order whether or not the tri-layer stacking induced metallic phase is destroyed by electron correlations? In order to reach a definite conclusion one would need to look at many different stacking arrangements.

(iv) The authors revised the part about the robustness of the Mott state against hydrostatic pressure. It is now more clear that this refers to the Mott state and not to the CCDW state.

In conclusion, I find that the revised version of the manuscript addresses most of the concerns raised in my previous report. However, I would appreciate if the authors could elaborate on the presentation of their STS data. In particular, as detailed above, I would like to see a comparison of the data to theoretical spatial resolved LDOS with different realistic smearing parameters. Given that the above comments are addressed, I would recommend the paper for publication.

Replies to the comments of the first reviewer

The authors of the manuscript answered questions and comments thoroughly. I found that all answers and modifications according to referee reports are more or less satisfactory. However, in some answers and in some revised sentences, there may be some issues to be discussed further.

Comment 1) In their reply 3, the authors argue that the topological networks or vortex structures are important because they are responsible for superconductivity. However, as shown in Ref. 13 and Ref. 38, the sample becomes a superconductor only with pressure and doping. With pressure, owing to its layered nature (large difference in in-plane and out-of-plane Young's modulus), TaS₂ should shrink a lot in direction perpendicular to the layer not in parallel direction. So, suitable theoretical consideration may use reduced interlayer distance or highly doped sample. For this situation, I wonder how this hexagonal network contribute to superconductivity with those conditions. It should be clearly written reason why the existing hexagonal metallic network in a pristine NCCDW sample contribute superconductivity only after pressure applied (Ref. 13) or high ionic gate doping (Ref. 38).

Reply 1) We appreciate this important comment. We agree that the TaS₂ sample becomes a superconductor only with pressure or doping and a proper consideration of those effects is necessary. In the manuscript, we already mentioned that the Mott insulating state is robust up to the in-plane lattice compression of 5 %. In the additional test calculations, the commensurate 1T-TaS₂ bulk was shown to maintain the insulating state up to the large reduction of interlayer spacing of about 0.4 Å (6.7% suppression) (see figure below). Moreover, the existence of the domain wall network itself under pressure has been well established by various previous experiments. The metallic nature of the

networked phase under pressure is out of question since the superconductivity requests it. Since the Mott insulating state is robust within the NCCDW phase (as shown here), the only source of the metallic electrons is the domain walls. Thus, the contribution of the domain wall electrons to the superconductivity becomes very much straightforward. Note also that the well-established phase diagram of the present system under pressure or doping obviously assumes the pressurized or doped phase for the superconductivity and the high temperature NCCDW phase are the same, seamlessly connected, phase without any experimental sign of changes between them. Thus, there is no clear reason to suspect that the metallic domain wall state can drastically change upon pressurizing or doping.

Figure. Total DOS of the on-top stacking 1T-TaS₂ bulk as a function of interlayer distance. The theoretical equilibrium distance is 5.992 Å.

Comment 2) Moreover, in Fig. 5(b), the bands of hexagonal network shows flat bands or very narrow bands width of few meV. The authors argue that those mini bands can fill the all the Mott gap and can contribute the superconductivity. However, considering disorders (distortion of hexagons etc) shown in NCCDW phase in Fig. 1a, the large scale hexagonal network seems not so much clean compared with twisted bilayer graphene and their small group velocity of energy bands may be altered a lot with disorders.

Reply 2) We agree that the disorder may in principle affect the details of the band structure. However, it is not true that the present domain wall network is less ordered than that of twisted bilayer graphene.

Figure. STM images of twisted bilayer graphene [Phys. Rev. Lett. **109**, 196802 (2012) and Sci. Rep. **6**, 27261 (2016)]

As shown in the above figure, one can also easily notice disorders in the twisted bilayer graphene [Phys. Rev. Lett. **109**, 196802 (2012) and Sci. Rep. **6**, 27261 (2016)]. See also the relevant theoretical consideration [PRM 2, 034004 (2018)]. On the other hand, our own FFT images of the NCCDW phase and various previous structure studies indicate a quite high degree of order for the domain wall network.

On the other hand, we also do not agree that the disorders will modify the band structures in the NCCDW network states more significantly than the twisted bilayer graphene because of the smaller group velocity (or, equivalently smaller bandwidth) of the network states. In fact, the overall bandwidth of the twisted bilayer graphene near the magic angle is about $O(10)$ meV, which is similar to that of the NCCDW network. One may still worry that disorders would alter the flat bands strongly. However, it has been known that the perturbative disorders do not localize the flat bands [Phys. Rev. B **82**, 104209 (2010)]. Hence, the flat bands and the associated effect on the manybody states may survive even after the inclusion of the disorders.

However, we agree that the effect of the imperfect hexagonal order in the band structure is an interesting issue requesting more through future theoretical study and thus we reflect this into the revised main text as following:

-While the Dirac physics of the conducting honeycomb network **and impact of disorders on the flat bands** [Phys. Rev. B **82**, 104209 (2010)] call for further investigation, the controllability on domain wall network suggested in recent experiments would provide exciting possibilities for engineered quantum states in this class of materials.

Also we add some discussion about the comparison of our network to the twisted bilayer graphene in the supplemental materials as following:

- Both the twisted bilayer graphene and our honeycomb network have weak disorders [Phys. Rev. Lett. **109**, 196802 (2012) and Sci. Rep. **6**, 27261 (2016)]. For example, there are some imperfect hexagons

in our network and imperfect triangles in twisted bilayer graphene. Naively one expects that such weak disorders would immediately localize the flat bands and completely destroy associated many-body physics. However, the previous study [Phys. Rev. B **82**, 104209 (2010)] surprisingly found that the flat bands do not get immediately localized but become critical. This implies that the flat bands are stronger against disorders than we naively expect. Though a more thorough investigation is desirable, we expect from the reference [Phys. Rev. B **82**, 104209 (2010)] that the flat bands retain relatively at spectrum even with the weak disorders and hence is expected to remain very susceptible to manybody physics.

Comment 3) As I wrote in my previous report, atomic vortex structures are interesting observations but their connections to other physics in TaS₂ are not directly supported by authors' experiment observations. Interesting band structures of hexagonal vortex network may be relevant with some observables in this compound also. But this also again lacks direct connections with superconductivity in doped or pressurized 1T-TaS₂.

Reply 3) As discussed above, we believe that there is no obvious reason to suspect the connection of the domain wall metallic states with the superconductivity in doped or pressurized samples. Denying this connection means that the high temperature NCCDW phase in the pristine sample and the NCCDW phase in doped or pressurized samples are different phases with different source of their metallicity. This statement denies the current phase diagram and has to introduce an unknown phase transition between these two NCCDW phase.

On the other hand, we agree that the role of vortex atomic structure may be not as clear as the hexagonal domain wall network. However, as we stated in our manuscript, the vortex structure provides substantial extra metallic electrons to the system, which would obviously help the emergence of the superconductivity.

In general, we appreciate the comments 1 and 3 of the reviewer, where he/she raised the concern that our present picture for the emerging superconductivity is relying on indirect evidence. We agree partly on this and toned down carefully our claims on the superconductivity and tried to explicitly state the limitation of the present argument throughout the manuscript.

Replies to the comments of the second reviewer

The authors kindly answered all questions raised in my previous report. More precisely, they

addressed the following points:

Comment 1) As also noted by the first Referee the original version of the manuscript did not discuss the role of interlayer coupling. The authors largely expanded this discussion in the revised version and also provide additional calculations and conclude that interlayer correlations are not relevant for the main result of their paper. Even though, I still think that interlayer coupling must play a significant role for the CCDW-NCCDW transition - simply because the stacking order of the CDW changes drastically from a disordered stacking in the CCDW to a well-ordered stacking in the NCCDW - I agree with the authors that, within their model, interlayer coupling is secondary for the metallization. However, the experimental verification of the DW metallization is still inconclusively (see also point (ii)).

Reply 1) We largely agree with the reviewer that the interlayer coupling may be important and the experimental verification of the DW metallization is still inconclusive. However, at present, the evidence itself that the NCCDW phase has a strong interlayer coupling is not clear enough. For example, a recent X-ray photoelectron spectroscopy study reported that the interlayer coupling is reduced in the NCCDW phase compared to the CCDW phase [Sci. Rep. **9**, 488 (2019)]. As given above for the comment 1 of the first reviewer, our extra calculation showed that the reduced interlayer spacing does not affect the Mott insulating state. Moreover, at present, it is not clear at all what kind of interlayer stacking order is present in NCCDW phase. (This point and the STS issue will be discussed below.)

Comment 2) I thank the authors for clarifying to me that the DW type 1 studied in the current paper is indeed distinct from the previously investigated DWs 2 and 4 - a very important piece of information that must have slipped my attention in the original manuscript.

I also appreciate the addition of STS data for the NC phase. Unfortunately, these data do not show much contrast between DW, domain and vortex. The authors attribute the absence of contrast to unavoidable thermal broadening at room temperature and show theoretical DOS with different Gaussian smearing. For values of the smearing parameter ~ 0.1 eV the theoretical spectra show a smeared out gap similar to the experimental data. However, 0.1 eV corresponds to about 1000K which is rather high compared to 300K.

Also, I would like to see the theoretical spatial resolved LDOS for DW, domain and vortex with different smearing parameters in order to verify that the contrast between domain, DW and vortex indeed gets wiped out.

Reply 2) We appreciate this important comment. However, the smearing parameter σ , which represents the electronic temperature without the lattice temperature, should not be directly compared with the real temperature of crystals [Sci. Rep. **5**, 16646 (2015)]. It gives only relative temperature scale. For example, a very high σ value of about 0.68 eV was needed to explain the phase transition of 120 K for 1T-TaSe₂ [Sci. Rep. **5**, 16646 (2015)]. The theoretical LDOS of the DW-1 is wiped out both for domain and domain wall regions at a smearing parameter of 0.15-0.2 eV. The fully systematic calculations for the domain, domain wall, and vortex DOS at different smearing parameters are not available yet but we can show a part of them. The revised SFig. 2(c) (shown below) shows the difference between domain and domain wall DOS at a smearing parameter of 0.15 and 0.2 eV. This difference almost disappears in the case of 0.2 eV in qualitative consistency with out STS spectra.

SFig. 2 STS spectra of the NC phase and theoretical DOS of DW-1 structure. (a) STS spectra, (b) Total DOS as a function of the electronic temperature parameter σ (eV). (c) Local DOS of domain and domain wall regions ($\sigma = 0.15$ and 0.2 eV).

Comment 3) I appreciate the expanded discussion about the interlayer coupling in the revised manuscript which gives a more balanced view of the debate whether or not strong electron-electron correlations are important for the low temperature phase of 1T-TaS₂. Nevertheless, I suppose that it strongly depends on the specific stacking and the imposed magnetic order whether or not the tri-layer stacking induced metallic phase is destroyed by electron correlations? In order to reach a definite conclusion one would need to look at many different stacking arrangements.

Reply 3) We appreciate this constructive comment. First of all, we point out that the magnetic order is not relevant here since no such order was experimentally observed down to a very low temperature [Ref. 24]. We agree that the specific stacking order may affect the electronic properties of 1T-TaS₂.

However, since the stacking order of the NCCDW phase is not specified yet, it is simply not possible to test every possible stacking orders. Moreover, one can easily estimate from the previous x-ray diffraction data [Ref. 46] that the degree of stacking order is very poor with a very weak and broad x-ray features, indicating a very short coherence length. With these given conditions, we can only deal with the most plausible stacking orders discussed in the previous works and two such stacking order models commonly support a crucial role of the metallization of domain walls. We believe that the referee would agree with our present approach.

Comment 4) The authors revised the part about the robustness of the Mott state against hydrostatic pressure. It is now more clear that this refers to the Mott state and not to the CCDW state.

In conclusion, I find that the revised version of the manuscript addresses most of the concerns raised in my previous report. However, I would appreciate if the authors could elaborate on the presentation of their STS data. In particular, as detailed above, I would like to see a comparison of the data to theoretical spatial resolved LDOS with different realistic smearing parameters. Given that the above comments are addressed, I would recommend the paper for publication.

Summary of changes

(1) We revised a sentence in discussion to tone down our claim.

=>

The novel metallic domain wall network with vortex charges in the NC-CDW phase, as revealed here, can be a key to the phase transition from Mott state to superconductivity.....

(2) We revised a sentence in discussion to mention our extra calculation for interlayer interaction.

=>

→ Our calculation indicates that the Mott state of C-CDW phase is robust against the in-plane (out-of-plane) lattice compression up to about 5 (6.7) %.

(3) We revised a sentence in discussion to mention the need of further investigation for interlayer

interaction.

=> While further investigation on the interlayer interaction is necessary, we note that there is.....

(4) We added a sentence in result to discuss the pressure or doping effect.

=>

The former is due to the honeycomb symmetry and the latter two produce huge density of states, which would amplify any small instability of the system and enhance the tendency toward many-body correlated states including superconductivity. Such enhancement appears when the Fermi level is tuned near one of the cascade of the flat bands, and the tuning may be achieved by applying pressure or doping [Refs 13, 22, 36, 37].

(5) We changed a sentence in result to discuss the disorder effect.

=>

While the Dirac physics of the conducting honeycomb network and impact of disorders on the flat bands [Phys. Rev. B **82**, 104209 (2010)] call for further investigation, the controllability on domain wall network suggested in recent experiments would provide exciting possibilities for engineered quantum states in this class of materials.

(6) We changed the SFig. 2 in Supplementary Materials to add the theoretical LDOS.

(7) We added a paragraph in Supplementary Materials to discuss the disorder effect.

Both the twisted bilayer graphene and our honeycomb network have weak disorders [Phys. Rev. Lett. **109**, 196802 (2012) and Sci. Rep. **6**, 27261 (2016)]. For example, there are some imperfect hexagons in our network and imperfect triangles in twisted bilayer graphene. Naively one expects that such weak disorders would immediately localize the flat bands and completely destroy associated many-body physics. However, the previous study [Phys. Rev. B **82**, 104209 (2010)] surprisingly found that the flat bands do not get immediately localized but become critical. This implies that the flat bands are stronger against disorders than we naively expect. Though a more thorough investigation is desirable, we expect from the reference [Phys. Rev. B **82**, 104209 (2010)] that the flat bands retain relatively at spectrum even with the weak disorders and hence is expected to remain very susceptible to many-body physics.

(8) We added three papers in reference.

- Phys. Rev. B **82**, 104209 (2010)

- Phys. Rev. Lett. **109**, 196802 (2012) in Supplementary Materials

- Sci. Rep. **6**, 27261 (2016) in Supplementary Materials

(9) We added an author, Jinwon Lee, who contributed in the STM(S) measurement and analysis.

Reviewers' comments:

Reviewer #1 (Remarks to the Author):

In this final response of the authors of NCOMMS-18-33392B, they answered the questions and comments of reviewers well and modified the paper accordingly. Therefore, I recommend a publication of the manuscript as it is.

Reviewer #2 (Remarks to the Author):

The authors addressed the comments of my previous report and added additional calculations.

However, I was very surprised when only a few days after I have submitted my previous report a new study, apparently coauthored by one of the authors (Doohee Cho) of the present manuscript, appeared in PRL [Lee et al., PRL 122, 106404 (2019)]. In line with previous studies, this PRL paper claims to clearly identify the CDW stacking, instead of the widely assumed Mott-Hubbard physics, as the driving mechanism behind the metal-insulator transition associated with the NCCDW-CCDW transition. Obviously, these findings are in stark contradiction with the manuscript at hand since the Mott mechanism - here modeled by DFT+U - forms the basis of the presented DFT model. In particular, since at least some of the authors are involved in both studies I am wondering why this contradiction is not discussed more thoroughly and why the authors do not cite the PRL paper. In the current version of the manuscript at hand the authors claim that: "While further investigation on the interlayer interaction seems necessary, we note that there is plenty of evidence that the intralayer effect is dominant for the low temperature physics", while at the same time one can read in the abstract of the PRL paper [Lee et al., PRL 122, 106404 (2019)]: "We show that the MIT is driven not by the 2D order itself, but by the vertical ordering of the 2D CDWs or the 3D CDW order".

I suggest the authors should discuss how to reconcile both studies.

Apart from this major concern, I would like to comment on reply 3 by the authors in which they state that previous x-ray data indicate a poor stacking order in the NCCDW phase: While this is true for the CCDW in which partial disorder yields broad super lattice reflections in the I-directions the NCCDW instead shows sharp superlattice reflections indicating a well-ordered stacking. The stacking order actually has been refined by Spijkerman et al. [Phys. Rev. B 56, 1357 (1997)].

Having said this, I still think that the current study about the discommensurations network in the NCCDW contains interesting results. However, in particular in the view of the above mentioned contradiction, I am not sure about the robustness and validity of the approach.

We feel sorry for an unexpected happening, the participation of one of our coauthor in the publication of a recent paper delivering a possibly contradictory message. In the following, we declare that the main authorship of the manuscript did not know such an overlapped authorship and, more importantly, the claim of this recent paper was already considered in our previous replies to the reviewer comments. Therefore, we do not find any obvious reason to prevent the prompt publication of the present manuscript.

- (1) The overlapped coauthor, Doohee Cho, participated in the STM experiment of the present manuscript and left the group 3 years ago (at a similar time with the other coauthors of the PRL paper mentioned by the reviewer). Thus, we had no chance to know the existence of and his own participation on the PRL paper, which was not mentioned by him. He endorsed his authorship and the content of the present manuscript properly before the manuscript was submitted. We contacted with him and heard that he joined the early discussion of the PRL paper and did not realize its possible conflict with the present manuscript.
- (2) The main point of the possible contradiction between the present work and the recent PRL paper is that the latter is explaining the metallic phase by a disordered stacking including a metallic stacking order (the L stacking in that paper). This particular stacking order is the simplest stacking sequence and is what was already considered in Supplementary Figs. 4 and 5b. Therein, we showed this stacking order is unstable against moderate electron correlation to fall into an insulating state and, in addition, does not affect the metallic property of the domain walls. Thus, the main difference of this paper and the current work is on the consideration of the electron correlation. It is very much unnatural to ignore fully the electron correlation of a very strongly localized d^1 electrons. These points were already discussed in page 4 of the manuscript (blue part in the present version). On that part of the manuscript, we explicitly cite this PRL paper (ref. [49]) together with other related references where we mention the possible effects of the interlayer ordering.

Going further, one can easily notice that the metallic L stacking phase cannot explain the experiments. First of all, in this phase the metallicity comes from the interlayer hopping while the experiments made clear the in-plane conductivity dominating as we pointed out in our previous replies. While this stacking order yields a strongly dispersing state in the in-plane band dispersion near Fermi level, multiple ARPES studies showed a flat band for the NCCDW phase. Moreover, the disordering transition picture suggested by the PRL paper is not consistent at all with the x-ray data for the CCDW and NCCDW phases as already

mentioned by the reviewer. The explicit criticism on this paper can be made in different occasions, focusing on details of the interlayer stacking and its temperature dependence.

Summary of the changes:

- (1) A new reference, ref. [49], was added and cited in page 4.

Reviewers' comments:

Reviewer #1 (Remarks to the Author):

The authors have successfully answered previous reviewers questions and comments. So, I recommend a publication of the manuscript as it is.

Reviewer #2 (Remarks to the Author):

The authors explained the circumstances that led to an obvious contradiction between the present manuscript and a recently published article in PRL with an overlap in authorship.

Although, I can understand these circumstances, I still find it a bit unexpected that a coauthor of two manuscripts which are submitted basically at the same time is unaware of this rather obvious conflict between the key results of both papers. It makes me wonder if the coauthors know the content of their papers - in my opinion a requirement in order to qualify as a coauthor.

However, my main concern is not so much about this formal issue. As stated previously, I rather doubt the robustness of the results of the current work. In their latest response, the authors pointed out that the main conceptual difference between the PRL paper and the current work lies in the treatment of electron correlations. Further, they say that it would be unnatural to neglect electron-electron correlations, which is correct. However, DFT+U is a rather crude approximation too, that comes with many pitfalls in particular for the (pretty extended) 5d states. To some extent the GGA or LDA XC-potential already contains (long-range) electron correlations which might be more appropriate for these extended 5d systems. At least the agreement between ARPES and DFT calculations including the stacking of the CCDW is exceptionally good (see the recent PRL paper and PRB 98, 195134 for instance). For the claim of the manuscript at hand this experimental support is widely missing or at least is not obvious to the reader. Most importantly, I think that using the correct (crystal) structure, which actually is 3D, is key in order to draw reliable conclusions.

Therefore, I believe that, although the results of this work about the discommensurations network are interesting, they are highly speculative and not well corroborated with experimental data. Accordingly, the derived conclusion is oversold and - in my opinion - does not meet the standards of nature communications but should be published in a more specialized journal.

Replies to the comments of the first reviewer

Comment:

The authors explained the circumstances that led to an obvious contradiction between the present manuscript and a recently published article in PRL with an overlap in authorship.

Although, I can understand these circumstances, I still find it a bit unexpected that a coauthor of two manuscripts which are submitted basically at the same time is unaware of this rather obvious conflict between the key results of both papers. It makes me wonder if the coauthors know the content of their papers - in my opinion a requirement in order to qualify as a coauthor.

Reply:

We authors apologize again the unexpected situation of the overlapped authorship with an apparently contradicting paper, which was published in the final stage of the review process. In this respect, we decided, with the full agreement of the overlapped author himself, Dr. Doohee Cho, that he will be excluded from the author list. As we already stated in the previous communication, he didn't realize fully the contradicting stories of the two publication with his names on. This fact itself seems to disqualify his own authorship of the present manuscript.

Comment:

However, my main concern is not so much about this formal issue. As stated previously, I rather doubt the robustness of the results of the current work. In their latest response, the authors pointed out that the main conceptual difference between the PRL paper and the current work lies in the treatment of electron correlations. Further, they say that it would be unnatural to neglect electron-electron correlations, which is correct. However, DFT+U is a rather crude approximation too, that comes with many pitfalls in particular for the (pretty extended) 5d states. To some extent the GGA or LDA XC-potential already contains (long-range) electron correlations which might be more appropriate for these extended 5d systems. At least the agreement between ARPES and DFT calculations including the stacking of the CCDW is exceptionally good (see the recent PRL paper and PRB 98, 195134 for instance). For the claim of the manuscript at hand this experimental support is widely missing or at least is not obvious to the reader. Most importantly, I think that using the correct (crystal) structure, which actually is 3D, is key in order to draw reliable conclusions.

Reply:

We largely agree on the general statement on the electron correlation of the reviewer and we believe that the reviewer more or less accepts our explanation (in our previous reply) on the limitation of the model of the recent PRL paper for the NCCDW phase and the metallization. This point was mentioned explicitly in the revised manuscript. The reviewer's concern now moves to the CCDW phase, questioning the robustness of the Mott insulator description of the present system. However, it is not true that the previous publications for the interlayer-coupled models (especially for the CCDW phase) is exceptionally well supported by the experimental data but the present model is not.

- (1) First of all, we restate that the stacking model predicted a band gap of 3 meV [PRB **98**, 195134 (2019)] which cannot explain the large band gap of the STS spectra. The GGA band gap was calculated to be 50 meV in PRL **122**, 106404 (2019), far from the experimental one (250-400 meV, see the figure below). It indicates that the extra (stronger) electron correlation has to be counted to explain the CCDW ground state.

PRB **92**, 085132 (2015)

PRB **98**, 195134 (2018)

- (2) The difference between the agreements of the present model and the previous stacking model for the ARPES in-plane dispersion is not substantial as shown below. The reviewer’s statement of “At least the agreement between ARPES and DFT calculations including the stacking of the CCDW is exceptionally good (see the recent PRL paper and PRB 98, 195134 for instance). For the claim of the manuscript at hand this experimental support is widely missing or at least is not obvious to the reader.” is thus not justified. This point was mentioned in the manuscript.

(Figure left) Experimental data of Nat. Commun. 3, 1069 (2012) together with our own calculation (single layer with U) for the in-plane band dispersion. (Right) The calculated in-plane dispersion of PRL with the guide to the eyes (while line).

PRL **122**, 106404 (2019)

(3) More importantly for the interlayer coupling, the out-of-plane dispersion is crucial to decide the proper structure. As shown below, our recent measurement (under preparation for publication) reveals the detailed k_z -dispersion of the topmost band. There can be two different interpretations (left and center) but one can notice that the k_z -dispersion is of the order of 50~120 meV in any case while both PRL 2019 and PRB 2018 predict a much large dispersion (more than 300 meV). Note that we have a full story to explain the extra splitting shown in the left figure (which is well out of the scope of the present topic) and the present conclusion for the metallization is not affected by this. In any case, this clearly shows that the current interlayer coupling models cannot reproduce the experimental data and the interlayer coupling energy is smaller than the energy scale of the Mott gap. This is what we consistently argued within our manuscript and the supplements and what matches with a huge body of experimental data acquired with different types of probes. If the reviewer think this information is essential for the discussion of the NCCDW phase, we may include this into the supplements.

(4) In fact, we can provide one more critical experimental data of our own, which tell that the stacking order or sequence does not affect the insulating property of the system at low temperature. The phase of CDW is abruptly shifted by the domain wall and the interlayer stacking order of domains in the opposite sides of the wall must be different. However, we showed that this different stacking order does not affect to the band gap, indicating the 2D Mott physics is more important than the stacking order at least at low temperature [Nat. Commun. **8**, 392 (2017)]. This point was explicitly stated in the revised manuscript.

Cho et al. Nat. Commun. 8 392 (2017)

(5) In conclusion, at the bottom line, we can clearly say that the current 3D stacking model is not conclusive at all and, at least, there are huge volume of experimental evidence that the 2D Mott physics plays crucial role at low temperature.

Comment:

Therefore, I believe that, although the results of this work about the discommensurations network are interesting, they are highly speculative and not well corroborated with experimental data. Accordingly, the derived conclusion is oversold and - in my opinion - does not meet the standards of nature communications but should be published in a more specialized journal. The authors of the manuscript answered questions and comments thoroughly. I found that all answers and modifications according to referee reports are more or less satisfactory. However, in some answers and in some revised sentences, there may be some issues to be discussed further.

Reply:

We appreciate some positive comments of the reviewer but do not agree that the present work oversells anything. The interlayer coupling issue is not related to overselling. However, we carefully reviewed our own manuscript again in this respect and revised our statements for the interlayer coupling issue.